# Flexible Hybrid Integration Hall Angle Sensor Compatible with the CMOS Process

**DOI:** 10.3390/s25030927

**Published:** 2025-02-04

**Authors:** Ye Luo, Youtong Fang, Yang Lv, Huaxiong Zheng, Ke Guan

**Affiliations:** 1College of Electrical Engineering, Zhejiang University, Hangzhou 310058, China; luoye30@csrzic.com (Y.L.); youtong@zju.edu.cn (Y.F.); 2Ningbo CRRC Times Transducer Technology Co., Ltd., Ningbo 315000, China; zhenghx@csrzic.com (H.Z.); guanke@csrzic.com (K.G.)

**Keywords:** Hall sensor, flexible sensor, ASIC, CMOS

## Abstract

Silicon-based Hall application-specific integrated circuit (ASIC) chips have become very successful, making them ideal for flexible electronic and sensor devices. In this study, we designed, simulated, and tested flexible hybrid integration angle sensors that can be made using complementary metal-oxide-semiconductor (CMOS) technology. These sensors are manufactured on a 100 µm-thick flexible polyimide (PI) membrane, which is suitable for large-scale production and has strong potential for industrial use. The Hall sensors have a sensitivity of 0.205 V/mT. Importantly, their sensitivity remains stable even after being bent to a minimum radius of 10 mm and after undergoing 100 bending cycles. The experiment shows that these flexible hybrid integration devices are promising as angle sensors.

## 1. Introduction

Hall sensors [1,2,3,4,5,6,7] are commonly used in automotive and consumer electronics [8] for accurately detecting positions and monitoring electric currents [9]. Currently, most Hall sensors rely on silicon complementary metal-oxide-semiconductor (CMOS) technology due to its established market presence and low production costs [10]. However, there is a growing demand for thin, flexible sensors [11,12,13,14,15] in emerging applications such as wearable devices [16,17,18,19], electronic skin [20,21], highly integrated robotic arms [22,23], and prosthetic limbs. Flexibility in these sensors not only enhances their mechanical properties but also allows them to fit into compact 3D structures [24,25]. This integration enables the addition of advanced sensing capabilities without altering the existing complex mechanical designs.

There are two main strategies for integrating flexible circuits: flexible hybrid electronics [26,27] and organic electronics [28,29]. Flexible hybrid electronics use flexible polymer substrates with rigid surface-mounted devices soldered onto them. In contrast, organic electronics are made from flexible organic semiconductor materials. Researchers are also focusing on improving the performance and miniaturizing organic transistors for all-organic integrated circuits (ICs) [30]. Additionally, flexible transistors using two-dimensional materials are becoming popular due to their thinness and high carrier mobility [31].

Directly making Hall materials on polymer substrates allows for flexible and high-performance devices [32], but there are challenges in integrating them on a large scale for functional use. Therefore, this paper focuses on developing a reliable flexible hybrid integration Hall sensor with an average thickness of approximately 100 μm, which can bend to a radius of at least 10 mm, and incorporates the advanced functions of traditional silicon CMOS Hall sensors.

## 2. Design and Fabrication

The core circuit of the flexible hybrid integration Hall sensor compatible with CMOS technology proposed in this study is a chip die fabricated using CMOS processes. The magnetic-sensitive element of the chip die is an N-type silicon epitaxial layer, and it includes a high-gain amplifier. The Hall voltage output, which is proportional to the magnetic induction strength in the *z*-axis direction, is delivered through pads on the chip, wire bonding, and copper lines on a flexible film. The fabrication process of this device is shown in Figure 1a. First, through a fabless model, a foundry is guided to produce the silicon-based Hall application-specific integrated circuit (ASIC) chip die, whose principles and circuit framework will be introduced later. Then, the substrate for the flexible device is created by sequentially spin-coating and curing a polyimide (PI) film, plating copper, applying a photoresist film, developing, and etching. The thickness of the PI layer is 100 μm. Next, a bonding adhesive is used to attach the Hall ASIC chip die to the designated position, and a wire bonding machine connects the copper lines on the substrate with the pads on the chip die. Finally, a layer of polydimethylsiloxane (PDMS) is applied and cured at the chip die location to protect the internal components.

The image of the device under test is shown in Figure 1b. To analyze the mechanical and electrical properties of the chip die under bending conditions, the chip die is arranged in the center of the area. The fabrication method is compatible with standard flexible printed circuit (FPC) designs, giving this device strong industrial production prospects. By measuring the output corresponding to different relative positions of the magnet and a single chip die, the lateral distance of the magnet can be determined. By modifying the circuit layout to arrange the chip die in an array and bending it for integration onto a curved surface, precise angle sensing can be achieved. The results of this sensing scheme will be discussed in the following sections. As shown in Figure 1c, the flexible substrate is connected to the chip die pads through gold wires with a micron-level radius. Although gold wires can withstand a certain amount of tensile strain, the connection between the wires and the pads is relatively weak. Therefore, the mechanical design focuses on ensuring the stability of electrical interconnections under complex conditions. Based on our manufacturing experience, the bonding point of the wire cannot withstand more than 4% of tensile or compressive strain.

Figure 1d shows a cross-section in the *z*-axis direction at the yellow dashed line from Figure 1c. Finite element simulation was used to analyze the solid mechanics of this cross-section, with results displayed in Figure 1e,f. Figure 1e shows the strain distribution in the xx direction at a bending radius of 10 mm, while Figure 1f shows the xx direction strain distribution inside the chip die at the red dashed line in Figure 1e. The thinner the chip die, the smaller the strain it experiences, and the presence of PDMS effectively ensures that the xx direction strain at the bonding point between the gold wire and the chip die pad remains below 4%. However, due to lower yield rates for chip die thicknesses of 150 μm or less, this study chooses a thickness of 200 μm for the chip die.

The Hall plate converts magnetic energy into electrical signals and generates the Hall voltage signal [33,34], which serves as the input to subsequent processing circuits. The Hall plate is a key component of the Hall ASIC chip, as shown in Figure 2a. To be compatible with CMOS processing, a silicon N epitaxial layer (Nepi) is typically used to generate the Hall voltage. There are two biasing methods for the Hall element: current biasing and voltage biasing [35,36]. Here, we choose the voltage biasing method.(1)$$V_{H}=\frac{w}{l}\mu \cdot V\cdot B_{z}=S_{V}\cdot V\cdot B_{z}$$

In this method, $V_{H}$ represents the Hall voltage, w and l are the width and length of the Hall element, μ is the carrier mobility of the material, V is the bias voltage, $B_{z}$ is the magnetic induction strength in the z-direction, and $S_{V}$ is the sensitivity.

Figure 2b shows the finite element analysis of the electric potential distribution of a 50 μm × 50 μm × 6 μm Hall plate under a 10 mT magnetic field, where a 4V voltage bias is applied to the A+ and A− terminals, and the Hall voltage is output at the B+ and B− terminals. As shown in Figure 2c, a larger bias voltage results in a greater output Hall voltage. This study selects a 4V bias voltage, with a sensitivity $S_{V}$ of 0.282 mV/mT. To reduce the offset voltage of the Hall plate, four Hall plates are typically placed symmetrically at the center, as illustrated in Figure 2d, while Figure 2e shows the SEM image of the Hall plate after removing the upper metal structures. Each Hall plate has four contact holes, two diagonally placed for providing the bias voltage and the other two for the positive and negative outputs of the Hall voltage. Figure 2f illustrates the connection relationship of the rotating current method, which controls four chopper switches with a clock signal. This ensures that during the positive and negative half-cycles, the Hall voltage is equal in magnitude but opposite in polarity, thereby eliminating the offset voltage associated with the Hall plate.

The framework of the Hall ASIC chip die designed in this paper is shown in Figure 3a. The core modules include the Hall plate, reference, logic section, chopper operational amplifier [37,38], demodulation filter, and buffer. This circuit has undergone comprehensive process voltage temperature (PVT) simulation tests, considering different operational states such as slow and fast power-on, over-voltage, over-current, and electrostatic discharge (ESD). It can provide magnetic sensing performance with sensitivity fluctuations of less than 3% over the temperature range of −40 to 125 °C. For confidentiality reasons, further details of the circuit structure are not provided here. The behavioral simulation of the Hall plate is specified using a Verilog-A model, allowing for the simulation of arbitrary input magnitudes of z-direction magnetic induction strength ($B_{z}$).

The upper part of Figure 3b shows the differential Hall voltage after chopper amplification, with an input of a sawtooth z-direction magnetic induction strength ($B_{z}$) ranging from −10 to 10 mT. The chopper switch divides the Hall voltage into positive and negative time periods. The upper part of Figure 3c shows the amplified differential chopper Hall voltage signal, with a chopper frequency of 1 MHz. Thanks to the appropriately sized switch settings and non-overlapping clocks, the chopper signal has minimal spikes, effectively enhancing the stability of the magnetic field sensing. The lower part of Figure 3b is the differential Hall voltage signal after demodulation filtering, which also shows minimal spikes due to the same favorable conditions.

Figure 3d presents the simulation of output signals at the VOUT and VREF terminals of the chip, with an input of sawtooth z-direction magnetic induction strength values ranging from −10 to 10 mT. The influence of the buffer and reference voltage leads to a differential Hall voltage signal oscillating around the VREF voltage. Typically, the VREF value is 2.5 V, enabling the measurement of the z-direction magnetic field strength at the chip’s location through the output value at VOUT. The simulation results indicate a sensing sensitivity of 0.21 V/mT for the chip. Figure 3e shows the chip layout after tape-out verification. To minimize strain interference on the Hall plate, it is placed in the center of the chip. The power supply voltage is arranged in a top-to-bottom order, while the signal transmission follows a right-to-left order. Figure 3f displays the chip after tape-out, demonstrating that this design process can be fully fabricated in a fabless mode, with broad market prospects [39].

## 3. Results and Discussion

Figure 4a shows the experimental setup for measuring magnetic field sensitivity. The magnetic field generator is a domestic solenoid coil that can produce a uniform magnetic field ranging from −50 to 50 mT by controlling the operating current. A digital gaussmeter is used to detect the magnitude of the magnetic induction strength. The flexible hybrid integration Hall sensor is mounted on a displacement stage, and its curvature is adjusted by moving the stage. A camera and curve fitting method are used to determine the bending radius at the position of the sensitive element being tested. The sensor is powered by connecting it to a 5V power supply via wires, and the output VOUT signal is measured using an oscilloscope. The actual testing setup is shown in Figure 4b, demonstrating that this direct wiring method provides strong environmental adaptability, allowing the flexible device to be easily installed on complex curved surfaces.

Figure 4c presents the magnetic sensing sensitivity test results before bending, during bending (with a bending radius of 10 mm), and after 100 bending cycles. Since the Young’s modulus of the silicon wafer (200 GPa) is much greater than that of PI and PDMS [40], the strain within the silicon wafer is relatively small. Tests show that a 10 mm bend has little effect on sensitivity, and multiple bends also do not significantly impact it, indicating that the structure has strong mechanical stability. The measured magnetic sensitivity is 0.205 V/mT, which is consistent with simulation results. The deviation is due to discrepancies between the Hall plate simulation and actual conditions, as well as process tolerances in fabrication.

Figure 4d shows the experimental setup for measuring the magnetic field response using a permanent magnet. As illustrated in Figure 4e, the output VOUT varies as the position of the permanent magnet changes relative to the flexible magnetic field sensor. However, since the direction of the magnetic field remains unchanged, the output values are all above 2.5 V.

We used a cylindrical permanent magnet for displacement experiments, ensuring that the magnet’s axis passed directly through the Hall sensor. The relationship between the magnetic flux density at the Hall sensor and the distance can be expressed by the following equation:(2)$$B\left(z\right)=\frac{B_{r}}{2}\left(\frac{z+T}{\sqrt{r^{2}+{\left(z+T\right)}^{2}}}-\frac{z}{\sqrt{r^{2}+z^{2}}}\right)$$

In this method, $B_{r}$ represents remanent magnetization of the magnet, T is the thickness of the magnet, z is the distance between the magnet surface and the chip die, and r is the radius of the cylindrical magnet. This characteristic allows for the development of current sensors, Hall distance sensors, and other devices. As shown in Figure 4f, when the permanent magnet moves horizontally, the magnetic sensitive element experiences a change in the magnetic field from negative to positive, resulting in corresponding variations in the output VOUT voltage. This property can be utilized to create angle sensors.

Figure 5a shows the flexible hybrid integration angle sensor, made up of multiple chip dies with mechanical and electrical properties similar to those previously described. Its flexible design allows for easy installation in locations where angle measurement is needed, requiring minimal space. As indicated in Figure 5b,c, the outer frame has a radius of 50 mm, which does not affect the sensor’s performance. A magnet is mounted on the inner shaft, and due to the sensor’s thin and highly sensitive nature, there is a selectable distance of 8 mm between the magnet and the side wall to ensure signal stability and strength. As the inner shaft rotates, the magnet also rotates, causing the VOUT signal values from the corresponding chip dies to change.

As shown in Figure 5d, the six chip dies can nearly cover a spatial angle of 30°, allowing for accurate angle measurements. Furthermore, by optimizing the circuit layout, a 360° angle can be covered using only a few chip dies. The flexible hybrid integration method lets us place sensing units exactly where needed in small spaces without changing the existing structure. This is helpful when the structure is already built. Additionally, using chip dies instead of packaged chips makes the design more flexible and smaller. This provides a convenient and reliable smart solution for existing aerospace devices, industrial robots, and other fields.

## 4. Conclusions

In summary, we report a flexible hybrid integration Hall sensor compatible with CMOS technology for magnetic field sensing, displacement sensing, and angle sensing. It can operate stably on complex and constrained surfaces and is easy to install. Mechanical analysis is used to design the structure to ensure stability in mechanical and electrical performance at a 10 mm bending radius. The flexible device’s magnetic field sensitivity is tested using a constructed magnetic field-testing platform, showing a sensitivity of 0.205 V/mT and confirming its mechanical stability. Finally, it demonstrates the application scenarios and operational status of the flexible angle sensor, which can nearly cover a spatial angle of 30° with just six dies. This CMOS-compatible flexible hybrid integration Hall sensor has broad application prospects in areas like robotic arms and advanced medical technology.

## Figures and Tables

**Figure 1 sensors-25-00927-f001:**
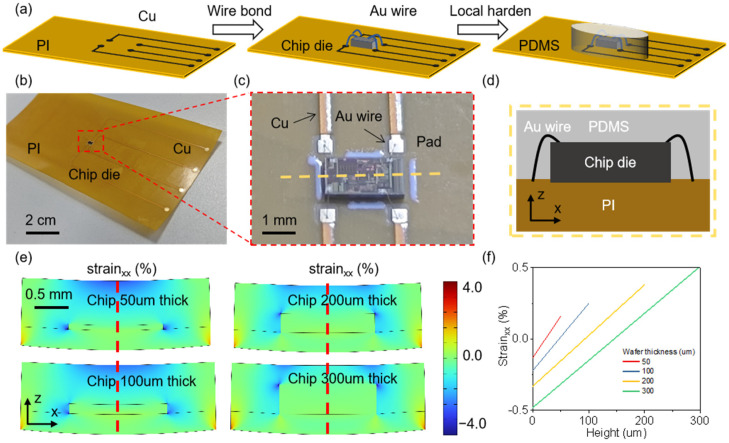
The preparation and mechanical analysis of the flexible hybrid integration Hall sensor. (**a**) The process for creating the flexible hybrid integration Hall sensor. (**b**) An experimental image of the finished sensor. (**c**) An image of the sensing unit. (**d**) A diagram of the electrical connections for the sensing unit. (**e**) The mechanical distribution in the xx direction across different thicknesses (50 µm, 100 µm, 200 µm, and 300 µm) at a bending radius of 10 mm. (**f**) The strain distribution in the silicon wafer along the dashed red line.

**Figure 2 sensors-25-00927-f002:**
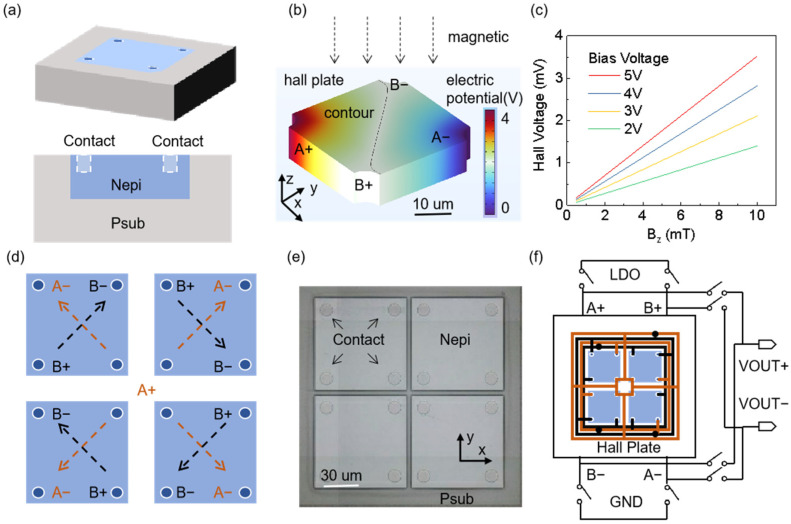
Analysis of the silicon-based Hall plate inside the chip die. (**a**) Schematic diagram of the 3D structure of the Hall plate; Nepi (N type silicon epitaxial layer); Psub (P type silicon substrate); PW (P type well). (**b**) Simulation of the electric potential distribution of the Hall plate under a magnetic field in the *z*-axis direction. (**c**) Simulation of Hall voltage values under different strengths of the *z*-axis magnetic field. (**d**) Schematic diagram of the electrical relationship of the rotating current method used to eliminate offset voltage. (**e**) SEM image of the Hall plate after removing the upper metal structures. (**f**) Electrical connection diagram of the rotating current method; LDO (Low dropout regulator).

**Figure 3 sensors-25-00927-f003:**
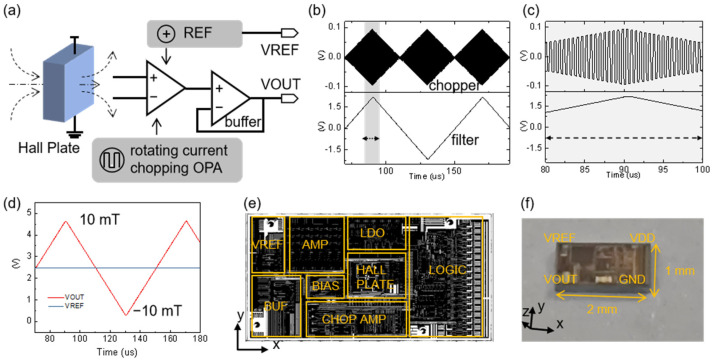
Circuit analysis of the chip die. (**a**) Schematic block diagram of the chip die circuit. (**b**,**c**) Simulation of the first-order chopper amplifier waveform and the demodulated filtered waveform. (**d**) Simulation of output values under different z-direction magnetic field strengths. (**e**) Layout of the chip die. (**f**) Image of the chip.

**Figure 4 sensors-25-00927-f004:**
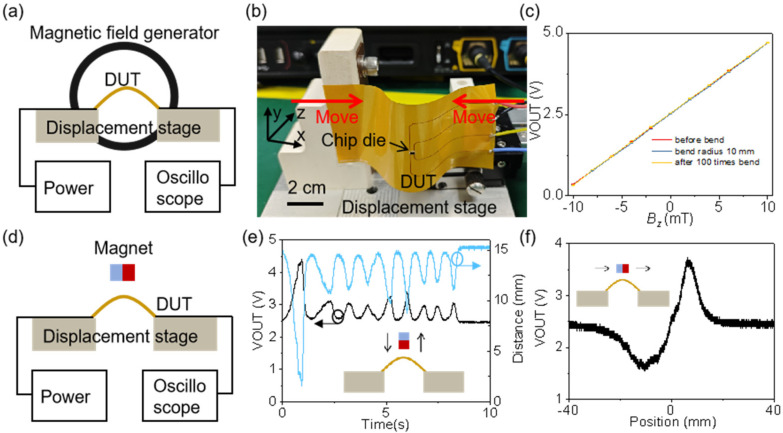
Experimental testing of the flexible hybrid integration Hall sensor. (**a**) Schematic diagram of the experimental setup for measuring magnetic field sensitivity. (**b**) Experimental testing image, which shows different bending radii of the flexible device achieved through planar buckling. (**c**) Measurement results of magnetic sensing sensitivity before bending, during bending (with a 10 mm bending radius), and after 100 cycles of bending. (**d**) Schematic diagram of the experimental setup for measuring magnetic field response. (**e**) Voltage test results with varying distances between the magnet and the sensitive element, the right *y*-axis represents the distance between the magnet surface and the chip die. (**f**) Voltage test results when the magnet moves laterally across the sensitive element.

**Figure 5 sensors-25-00927-f005:**
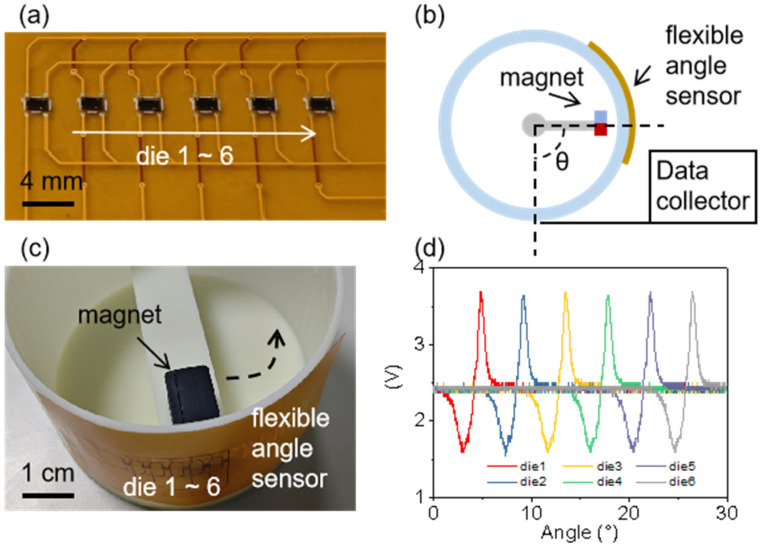
Flexible hybrid integration angle sensor. (**a**) Photo of the flexible hybrid integration angle sensor. (**b**) Schematic diagram of the flexible hybrid integration angle sensor device, which is directly attached to the outer edge of the measurement location, with the magnet mounted on an internal rotating shaft to collect voltage data through the circuit. (**c**) Photo of the device. (**d**) Voltage output at different rotation positions of the magnet.

## Data Availability

The data presented in this study are available upon request.

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
