# Peer review of "Flexible Hybrid Integration Hall Angle Sensor Compatible with the CMOS Process"

_sensors, 2025, doi:10.3390/s25030927_

Round 1
Reviewer 1 Report
Comments and Suggestions for Authors
Please find the attached detailed report.

Author Response
Dear Editor and Reviewers, please see the attachment for my response.

Reviewer 2 Report
Comments and Suggestions for Authors
In this study, we designed, simulated, and tested flexible angle sensors that can be made using CMOS technology. These sensors are man-ufactured on a 100 µm thick flexible PI membrane, which is suitable for large-scale pro-duction and has strong potential for industrial use. The Hall sensors have a sensitivity of 0.205V/mT. Importantly, their sensitivity remains stable even after being bent to a mini- mum radius of 10 mm and after undergoing 100 bending cycles. The experiment shows that these flexible devices are promising as angle sensors.
1. The introduction is too thin and rich in suggestions.
2. Some simulation results in this paper are fuzzy.
3. It is mentioned that placing Hall plate in the center of the chip can minimize strain interference, but there is a lack of in-depth analysis on the specific mechanism and influence of strain interference.
4. In this paper, it is introduced that there are two biasing methods for Hall elements: current bias and voltage bias. Why do we choose the voltage bias method instead of the current bias method?
5. The conclusion is too long, so it is suggested that the author simplify the conclusion and highlight the main work and contribution of the paper.
Author Response

(The authors gave the same response as above.)

Round 2
Reviewer 1 Report
Comments and Suggestions for Authors
Great work and congrats!
Author Response
Dear Reviewer,
Thank you very much for your review and guidance on our manuscript. Your professional opinions and suggestions have been crucial in improving the quality and depth of our paper. By carefully considering each of your feedback points and revising accordingly, I believe the manuscript will become more rigorous and complete. Throughout this revision process, your valuable experience and insights have greatly inspired and helped us.
Once again, thank you for the time and effort you have devoted to our research. We sincerely appreciate it.
Best regards
Reviewer 2 Report
Comments and Suggestions for Authors
1. Why is the thickness of the Hall sensor 100 microns mentioned in the Introduction section, but the chip used in the experiment is 200 microns?
2. When introducing precise angle sensing in Design and Fabircation, it is recommended to refer to the corresponding chapter.
Author Response
Dear Reviewer,
Please see the attachment for the response.
